# OpenReview forum: "SSL4RL:  Revisiting Self-supervised Learning as Intrinsic Reward for Visual-Language Reasoning"
_ICLR.cc/2026/Conference — ICLR 2026 Conference Desk Rejected Submission_

### Official Review · Reviewer_5KPi · 2025-10-18

**Soundness:** 3
**Presentation:** 3
**Contribution:** 3
**Rating:** 6
**Confidence:** 3

**Summary:**

SSL4RL reframes classic self-supervised tasks (rotation, jigsaw, contrastive, position) as verifiable intrinsic rewards for RL post-training, using GRPO to improve visual grounding and reasoning without human judges. It achieves significant gains on MMBench, SEED-Bench, and ImageNet-1K, identifies key design factors (task difficulty, model scale, semantic alignment), and generalizes to graph learning.

**Strengths:**

- Well written with clear and smooth logic.
- The method is novel, with rich experiments validated across different tasks.
- The analysis is comprehensive, including relationships between performance and task difficulty, as well as model size.

**Weaknesses:**

- What is the method’s performance upper bound, and how does it scale with data? At present, the achievable ceiling appears limited.
- The experiments are not sufficient: the VLM benchmarks include only MMBench and Seed-Bench, and the validation uses only the Qwen-2.5-VL series. It would be better to verify effectiveness on other VLMs and benchmarks.
- The tables in the main text detail category-wise performance on MMBench and Seed-Bench, but there is no analysis of how different self-supervised tasks impact performance on different benchmark sub-tasks.

**Questions:**

- Characterize the relationship between performance and data scaling: how does the method’s effectiveness evolve as data volume increases?
- Validate on additional benchmarks and train models beyond the Qwen-2.5-VL series to assess generality.
- Analyze how different self-supervised tasks affect performance across distinct benchmark sub-tasks.

---

> ### Author Response · Authors · 2025-11-24
>
> We thank Reviewer 5KPi for the positive feedback on the writing clarity, methodological novelty, comprehensive experiments, and in-depth analysis. The thoughtful questions further sharpen our focus, and we address your concerns as follows.
>
> ---
>
> **Q1.** Characterize the relationship between performance and data scaling: how does the method’s effectiveness evolve as data volume increases?
>
> **A1.** To characterize the relationship between performance and data scaling for SSL4RL, we progressively expand our training dataset and evaluate the resulting models on the MMBench benchmark, obtaining three training regimes with increasing data volumes:
> - Base Set: ~4,000 samples from MMBench.
> - Extended Set: ~18,000 samples from a mixture of MMBench and SEED-Bench.
> - Full Set: ~118,000 samples from MMBench, SEED-Bench, and ImageNet.
>
> Under time constraints, we select the Position task as the representative SSL4RL method for evaluation. The results, detailed in ***Table R1***, demonstrate a clear positive scaling relationship across almost all subtasks. **When scaling from Base Set to Extended Set, the average performance on MMBench improved from 80.08% to 81.38%, a gain of +1.30%. A further expansion to the Full Set yielded an additional +1.04% improvement, reaching 82.42%.** This consistent, monotonic improvement suggests that the effectiveness of SSL4RL continues to benefit from larger, more diverse datasets. **The performance gains are not uniform across all sub-categories, which provides deeper insight.** For instance, the most significant improvements are observed in Cross-Instance Fine-grained Perception, which saw a substantial jump of **6.36%**  from the smallest to the largest dataset (66.51% vs 72.87%). This indicates that tasks requiring nuanced comparisons across different images benefit from exposure to a broader visual world. Conversely, capabilities like Relation and Attribute Reasoning showed more modest gains, potentially plateauing earlier.
>
> In summary, our analysis confirms that SSL4RL effectively translates increased data volume into enhanced performance. The experiments indicate the potential for further gains through even more extensive pre-training, establishing SSL4RL as a promising and scalable strategy for vision-language model alignment.
>
> **Table R1. Performance (%) of SSL4RL models on MMBench benchmark with different training data scales.**
> | Model | Train Volumes | Logical | Relation | Attribute | Coarse | Cross-Inst. | Single-Inst. | Average |
> | - | - | - | - | - | - | - | - | - |
> | Qwen2.5-VL-3B | - |  61.77 | 41.54 | 76.62 | 73.55 | 64.32 | 82.06 | 72.99 |
> | SSL4RL-Position | 4,000 | 67.65 | 77.19 | 82.22 | 82.15 | 66.51 | 85.39 | 80.08 |
> | ~ | 18,000 | *68.71* | *79.26* | **85.03** | *81.65* | *70.16* | *86.47* | *81.38* |
> | ~ | 118,000 | **68.77** | **79.29** | *84.26* | **83.66** | **72.87** | **88.31** | **82.42** |
>
> **NOTE for Revised PDF: The experiments above are added to Section 4.5.1 and discussed in Appendix E in detail.**
>
> ---

---

> ### Author Response · Authors · 2025-11-24
>
> **Q2.** Validate on additional benchmarks and train models beyond the Qwen-2.5-VL series to assess generality.
>
> **A2.** **For benchmarks,** besides MMBench and SEED-Bench, we further consider four representative VQA benchmarks, **V\* [1], RealWorldQA [2], BLINK [3], MME-RealWorld-Lite [4]**, featuring challenging current VLMs' ability to capture image details, spatial understanding, visual perception, real-world applications, and so on. The detailed descriptions of benchmarks are provided in **Appendix I.2**.
>
> We consider the four SSL4RL strategies: Rotation, Jigsaw, Contrastive, and Position, under the same training and evaluation settings in Section 4.1. The results consistently demonstrate the effectiveness of SSL4RL. **For the 3B models (*Table R2*), our method provides a substantial average performance gain of 7.27% over the base model.** The improvements are particularly pronounced on V* (+8.90%) and RealWorldQA (+9.55%), underscoring that SSL4RL significantly enhances the model's ability to process fine-grained details in complex, real-world images. These consistent results across a diverse set of benchmarks verify the effectiveness of the proposed SSL4RL framework.
>
> **Table R2. Test performance (%) of SSL4RL models on extended VQA benchmarks.**
> | Category | Model | V* | RealWorldQA | BLINK | MME-RealWorld-Lite | Average |
> | - | - | - | - | - | - | - |
> | Base | Qwen2.5-VL-3B | 59.16 | 52.67 | 42.13 | 32.41 | 45.85 |
> | SSL4RL | Rotation | 62.30 | *62.09* | **48.18** | 34.18 | *51.68* |
> | ~ | Jigsaw | *63.35* | **62.22** | 45.18 | *35.12* | 51.46 |
> | ~ | Contrastive | 60.20 | 58.03 | 45.13 | 30.17 | 48.38 |
> | ~ | Position | **68.06** | 59.86 | *46.39* | **38.19** | **53.12** |
>
> [1]. Wu, Penghao, and Saining Xie. "V*: Guided visual search as a core mechanism in multimodal llms." CVPR. 2024.
>
> [2].  Team, xAI, et al. "Grok-1.5 Vision Preview" 2024.
>
> [3]. Fu, Xingyu, et al. "Blink: Multimodal large language models can see but not perceive." ECCV, 2024.
>
> [4]. Zhang, Yi-Fan, et al. "Mme-realworld: Could your multimodal llm challenge high-resolution real-world scenarios that are difficult for humans?." arXiv:2408.13257 (2024).
>
> **NOTE for Revised PDF: The above experiments are added to Section 4.1 and discussed in Appendix I.1 in detail.**

---

> ### Author Response · Authors · 2025-11-24
>
> *(Continues for A2)*
>
> **For base models**, we further select **Gemma3-4B** [5], a recent and powerful model from Google with a distinct architecture. To ensure a fair comparison under computational constraints, we adapted our training setup by halving the batch size to 256 while meticulously preserving all other hyperparameters and training procedures. Results presented in ***Table R3*** provide strong evidence for the general applicability of our method. **On the Gemma3 base model, our SSL4RL models yield a consistent and notable average performance improvement of 2.88% on MMBench. The gain is particularly pronounced in Cross-instance Perception (5.76%), Logical Reasoning (4.35%), and Attribute Reasoning (4.13%).**  Moreover, **the relative efficacy of the individual SSL4RL tasks is remarkably consistent with our prior findings.** As with Qwen2.5-VL, tasks like Rotation and Position confer the most significant benefits, while the simpler Contrastive task shows more modest gains. The consistent results validate that SSL4RL is a general-purpose principle for VLMs, not an artifact of a particular model family.
>
> **Table R3. Test performance (%) on MMBench downstream tasks. The base model is Gemma3-4B.**
> | Category | Model | Logical | Relation | Attribute | Coarse | Cross-Inst. | Single-Inst. | Average |
> | - | - | - | - | - | - | - | - | - |
> | Base | Gemma3-4B | 63.12 | 80.56 | 83.24 | 80.43 | 68.65 | 78.85 | 78.30 |
> | SSL4RL | Rotation | **67.47** | 81.16 | 86.12 | **83.27** | 72.35 | **82.09** | **81.38** |
> | ~ | Jigsaw | 63.77 | **84.39** | *86.69* | 82.99 | *73.72* | 80.67 | 81.10 |
> | ~ | Contrastive | 63.89 | 81.94 | **87.37** | 82.30 | 71.79 | *81.70* | 80.75 |
> | ~ | Position | *64.81* | *82.26* | 86.54 | *83.08* | **74.41** | 80.89 | *81.15* |
>
> [5]. Team, Gemma, et al. "Gemma 3 technical report." arXiv:2503.19786 (2025).
>
> **NOTE for Revised PDF: The experiments are added to Section 4.5.3 and discussed in Appendix H in detail.**

---

> ### Author Response · Authors · 2025-11-24
>
> **Q3.** Analyze how different self-supervised tasks affect performance across distinct benchmark sub-tasks.
>
> **A3.** Through a sub-task specific analysis, we find that **Rotation** exhibits superior performance in Relation Reasoning and Cross-Instance Perception, which can be attributed to its inherent requirement for a structural comprehension of object orientation and spatial relationships. **Position** demonstrates leading results in Logical Reasoning and holistic Scene Understanding, as its objective of reconstructing a coherent scene from disparate patches fosters robust integrative scene modeling. **Jigsaw** shows consistent utility in domains reliant on contextual reasoning. Limited by the insufficient augmentations, **Contrastive**'s optimization for global image similarity appears to inadequately cultivate the fine-grained and relational reasoning capabilities. Taken together, the detailed analysis provides insights into how different SSL4RL objectives function for distinct downstream tasks.
>
> **NOTE for Revised PDF: The discussions are added to the Analysis part in Section 4.1.**
>
> ---
>
> Thanks for your comments and hope our answers could address your concerns. Please let us know if there is more to clarify. We are happy to address them during the discussion stage.

---

> > ### Comment · Reviewer_5KPi · 2025-11-24
> >
> > Thank you for your detailed and thoughtful response.
> >
> > Overall, I view the paper as marginally above the acceptance threshold. I am therefore keeping my rating unchanged, but I have increased my confidence in this assessment.

---

### Official Review · Reviewer_Sy7L · 2025-11-01

**Soundness:** 3
**Presentation:** 3
**Contribution:** 3
**Rating:** 6
**Confidence:** 4

**Summary:**

The authors propose a method that uses the verifiability of self-supervised learning tasks to enable RLVR on LVLMs without other labels. The method is interesting and clever.

**Strengths:**

1. The method is clever.
2. he writing is clear, experiments are solid.

**Weaknesses:**

1. The tested benchmarks are somewhat limited. It's unclear how this method affects the model's multimodal alignment and visual reasoning.
2. Only tested on the Qwen-VL architecture, which limits the generalizability of the method.

**Questions:**

Obviously, this method may enhance the model's robustness. Have the authors evaluated this type of metric?

---

> ### Author Response · Authors · 2025-11-24
>
> We thank Reviewer Sy7L for the positive feedback on our method's cleverness, the clear writing, and the solid experiments. The comments are instrumental in helping us strengthen the paper, which we summarize and address as below.
>
> ----
>
> **Q1.** The tested benchmarks are somewhat limited. It's unclear how this method affects the model's multimodal alignment and visual reasoning.
>
> **A1.** Besides MMBench and SEED-Bench, we further consider an image-captioning task for multimodal alignment and four VQA benchmarks for visual reasoning.
>
> **For visual reasoning,** we select four representative VQA benchmarks, **V\* [1], RealWorldQA [2], BLINK [3], MME-RealWorld-Lite [4]**, featuring challenging current VLMs' ability of visual reasoning, spatial understanding, visual perception, real-world applications, and so on. The detailed descriptions of benchmarks are provided in **Appendix I.2**.
>
> We consider the four SSL4RL strategies: Rotation, Jigsaw, Contrastive, and Position, under the same training and evaluation settings in Section 4.1. The results consistently demonstrate the effectiveness of SSL4RL. **For the 3B models (*Table R1*), our method provides a substantial average performance gain of 7.27% over the base model.** The improvements are particularly pronounced on V* (+8.90%) and RealWorldQA (+9.55%), underscoring that SSL4RL significantly enhances the model's ability to process fine-grained details in complex, real-world images. These consistent results across a diverse set of benchmarks verify the effectiveness of the proposed SSL4RL framework.
>
> **Table R1. Test performance (%) of SSL4RL models on extended VQA benchmarks.**
> | Category | Model | V* | RealWorldQA | BLINK | MME-RealWorld-Lite | Average |
> | - | - | - | - | - | - | - |
> | Base | Qwen2.5-VL-3B | 59.16 | 52.67 | 42.13 | 32.41 | 45.85 |
> | SSL4RL | Rotation | 62.30 | *62.09* | **48.18** | 34.18 | *51.68* |
> | ~ | Jigsaw | *63.35* | **62.22** | 45.18 | *35.12* | 51.46 |
> | ~ | Contrastive | 60.20 | 58.03 | 45.13 | 30.17 | 48.38 |
> | ~ | Position | **68.06** | 59.86 | *46.39* | **38.19** | **53.12** |
>
> [1]. Wu, Penghao, and Saining Xie. "V*: Guided visual search as a core mechanism in multimodal llms." CVPR. 2024.
>
> [2]. x.ai Team. "Grok-1.5 Vision Preview" 2024.
>
> [3]. Fu, Xingyu, et al. "Blink: Multimodal large language models can see but not perceive." ECCV, 2024.
>
> [4]. Zhang, Yi-Fan, et al. "Mme-realworld: Could your multimodal llm challenge high-resolution real-world scenarios that are difficult for humans?." arXiv:2408.13257 (2024).
>
> **NOTE for Revised PDF: The above experiments are added to Section 4.1 and discussed in Appendix I.1 in detail.**

---

> ### Author Response · Authors · 2025-11-24
>
> *(Continues for A1)*
>
> **For multimodal captioning,** we leverage **a recent image captioning platform, CapArena [4]**, which contains over 6,000 human-annotated pairwise preference battles. In CapArena, captions from a test model are compared against those from strong baseline models (GPT-4o, CogVLM-19B, or MiniCPM-8B) using a LLM (GPT-4o) as a judge, with human references provided for context. The winner is assigned +1, the loser with -1, and 0 for a draw in each pairwise comparison. This challenging task mirrors the real-world challenge of improving a model without a single clear and correct answer.
>
> We evaluate our four SSL4RL strategies on CapArena against the base Qwen2.5-VL-3B model. For easier comparison, we present the final scores after min-max normalization in ***Table R2***. **SSL4RL consistently improves over the base model, with the largest performance gain of 8.14 points (56.45 vs. 48.31).** This clearly shows that the self-supervised rewards provide a meaningful learning signal even in the absence of a verifiable ground truth.
>
> **Qualitative Insights.** We conduct a qualitative comparison in **Appendix Table 27**, shedding light on how SSL4RL enhances open-ended generation:
> - **Enhanced Detail Capture.** SSL4RL models correctly capture more details in the image. For instance, our SSL4RL model correctly identifies a "scoreboard" displaying "1 0 0" and a "disabled persons' sign," which the base model either misinterprets or omits.
> - **Improved Spatial Reasoning.** SSL4RL models tend to use more precise spatial descriptors like "behind the fence", "to the left of", and "the center of the court", aligning well with the spatial reasoning ability required by SSL pretext tasks.
>
> In summary, this experiment demonstrates that SSL4RL is not limited to tasks with easily verifiable rewards. By providing a dense, automated learning signal derived from the data's intrinsic structure, our method successfully improves model performance on the complex, open-ended task of image captioning.
>
> **Table R2. Performance of SSL4RL models on CapArena Platform. GPT-Score: the score compared with GPT-4o. Cog-Score: the score compared with CogVLM-19B. CPM-Score: the score compared with MiniCPM-8B. Average: the average score.**
> | Category | Model | GPT-Score | Cog-Score | CPM-Score | Average |
> | - | - | - | - | - | - |
> | Base | Qwen2.5-VL-3B | 0.00 | 6.48 | 92.96 | 48.31 |
> | SSL4RL | Rotation | **19.15** | 49.30 | 96.48 | 55.28 |
> | ~ | Jigsaw | *8.87* | 54.93 | *98.59* | *55.64* |
> | ~ | Contrastive | 4.96 | *57.95* | **100.00** | **56.45** |
> | ~ | Position | 0.00 | **59.15** | 92.25 | 51.04 |
>
> [4] Cheng, Kanzhi, et al. "Caparena: Benchmarking and analyzing detailed image captioning in the llm era." arXiv:2503.12329 (2025).
>
> **NOTE for Revised PDF: The above experiments are added to Section 4.2 and discussed in Appendix I.2 in detail.**

---

> ### Author Response · Authors · 2025-11-24
>
> **Q2.** Only tested on the Qwen-VL architecture, which limits the generalizability of the method.
>
> **A2.** To evaluate whether the benefits of SSL4RL extend to architectures beyond, we select **Gemma3-4B** [5], a recent and powerful model from Google with a distinct architecture. Under computational constraints, we adapted our training setup by halving the batch size to 256 while meticulously preserving all other hyperparameters and training procedures. Results presented in ***Table R3*** provide strong evidence for the general applicability of our method. **On the Gemma3 base model, our SSL4RL models yield a consistent and notable average performance improvement of 2.88% on MMBench. The gain is particularly pronounced in Cross-instance Perception (5.76%), Logical Reasoning (4.35%), and Attribute Reasoning (4.13%).** Moreover, **the relative efficacy of the individual SSL4RL tasks is remarkably consistent with our prior findings.** As with Qwen2.5-VL, tasks like Rotation and Position confer the most significant benefits, while the simpler Contrastive task shows more modest gains. The consistent results validate that SSL4RL is a general-purpose principle for VLMs, not an artifact of a particular model family.
>
> **Table R3. Test performance (%) on MMBench downstream tasks. The base model is Gemma3-4B.**
> | Category | Model | Logical | Relation | Attribute | Coarse | Cross-Inst. | Single-Inst. | Average |
> | - | - | - | - | - | - | - | - | - |
> | Base | Gemma3-4B | 63.12 | 80.56 | 83.24 | 80.43 | 68.65 | 78.85 | 78.30 |
> | SSL4RL | Rotation | **67.47** | 81.16 | 86.12 | **83.27** | 72.35 | **82.09** | **81.38** |
> | ~ | Jigsaw | 63.77 | **84.39** | *86.69* | 82.99 | *73.72* | 80.67 | 81.10 |
> | ~ | Contrastive | 63.89 | 81.94 | **87.37** | 82.30 | 71.79 | *81.70* | 80.75 |
> | ~ | Position | *64.81* | *82.26* | 86.54 | *83.08* | **74.41** | 80.89 | *81.15* |
>
> [5]. Team, Gemma, et al. "Gemma 3 technical report." arXiv:2503.19786 (2025).
>
> **NOTE for Revised PDF: The experiments are added to Section 4.5.3 and discussed in Appendix H in detail.**

---

> ### Author Response · Authors · 2025-11-24
>
> **Q3.** Obviously, this method may enhance the model's robustness. Have the authors evaluated this type of metric?
>
> **A3.** Thanks for this constructive question. To evaluate our model's robustness, we conduct a comprehensive evaluation under various image perturbations. We design two levels of perturbation—weak and strong—to assess the models' resilience to visual corruptions. The weak perturbation applies a series of common image transformations, each with a probability of 0.5, including color jittering, random conversion to grayscale, Gaussian blur, and horizontal flipping. The strong perturbation builds upon the weak one by incorporating an additional multi-crop strategy. This strategy generates two 224×224 pixel crops per image via randomly resized cropping (scale range [0.08, 1.0]), presenting a more significant challenge to the model's perception. **Examples of these perturbed images are shown in Appendix Figure 6.**
>
> We evaluate our SSL4RL models and the base model on the perturbed MMBench benchmarks. **As shown in *Table R4* and *Table R5*, our method consistently demonstrates superior robustness across both perturbation levels.** For instance, under weak perturbation, our SSL4RL-Rotation model achieves an average score of 72.11%, outperforming the base model at 66.80%. This performance gap is maintained under strong perturbation, where our SSL4RL-Position model scores 64.37% compared to the base model's 59.62%. These results confirm that our SSL4RL strategy effectively enhances the model's robustness to photometric variations.
>
> **Table R4. Test performance (%) of SSL4RL models on the weakly perturbed MMBench benchmark.**
> | Category | Model | Logical | Relation | Attribute | Coarse | Cross-Inst. | Single-Inst. | Average |
> | - | - | - | - | - | - | - | - | - |
> | Base model | Qwen2.5-VL-3B | 49.58 | 45.06 | 76.94 | 67.80 | 59.33 | 68.46 | 66.80 |
> | SSL4RL | Rotation | **61.00** | **72.87** | 78.66 | *70.18* | **62.57** | *72.89* | **72.11** |
> | ~ | Jigsaw | 55.47 | 70.80 | **79.38** | **71.23** | 56.28 | 71.84 | 70.84 |
> | ~ | Contrastive | 54.46 | 62.05 | 67.33 | 55.80 | 47.64 | 64.02 | 59.94 |
> | ~ | Position | *57.97* | *72.77* | *79.01* | 70.40 | *57.55* | **73.39** | *71.70* |
>
> **Table R5. Test performance (%) of SSL4RL models on strongly perturbed MMBench benchmark.**
> | Category | Model | Logical | Relation | Attribute | Coarse | Cross-Inst. | Single-Inst. | Average |
> | - | - | - | - | - | - | - | - | - |
> | Base model | Qwen2.5-VL-3B | 45.08 | 44.99 | 71.65 | 59.88 | 53.89 | 58.62 | 59.62 |
> | SSL4RL | Rotation | *52.69* | *60.47* | 70.96 | *64.33* | **55.04** | *62.51* | *63.38* |
> | ~ | Jigsaw | 46.45 | 59.62 | *72.86* | 63.90 | 52.40 | 62.47 | 62.00 |
> | ~ | Contrastive | 45.56 | 46.36 | 60.86 | 49.50 | 42.14 | 53.20 | 51.25 |
> | ~ | Position | **53.43** | **61.37** | **72.95** | **65.85** | *53.68* | **65.12** | **64.37** |
>
>
> **NOTE for Revised PDF: The above experiments are added to Section 4.4 and discussed in Appendix F in detail.**
>
> ---
>
> We appreciate your time in reviewing our work and hope these responses resolve your queries. Please let us know if there are any other points you wish to discuss.

---

### Official Review · Reviewer_1F8A · 2025-11-05

**Soundness:** 2
**Presentation:** 3
**Contribution:** 2
**Rating:** 4
**Confidence:** 3

**Summary:**

This paper reframes classic self-supervised tasks (rotation, jigsaw, contrastive, patch position) as verifiable rewards for RL training of VLMs and optimizes them with GRPO. Mainly on 3B level models, it reports consistent gains on MMBench and SEED-Bench. Ablations reveal that task difficulty must match model capacity, gains diminish at 7B scale, and naive multi-reward mixing does not help.

**Strengths:**

1. The idea of using SSL tasks as verifiable RL rewards is simple and practical, avoiding noisy LLM-as-judge signals, however, it can only apply to certain few tasks (e.g., image captioning is not feasible).
2. The experimental spans reasoning benchmarks, ImageNet, and even graph tasks, showing some cross-domain generality.
3. The presented qualitative attention maps and error analyses help connect the method to observed reductions in language priors.

**Weaknesses:**

1. The baseline coverage is limited: there are no direct comparisons with strong RLHF/DPO or verifier-driven multimodal RL training pipelines, making it hard to judge competitiveness. Also, the improvements on some cases are very marginal.
2. The 7B results show small gains, supporting a capacity–difficulty mismatch; the method’s effectiveness at modern larger scales remains unclear without harder SSL tasks.

3. Since the RL training is often unstable, it would be good to see the training dynamics such as loss curve and entropy.
4. Can the authors provide some insights into not-easy-to-verify tasks, since the chosen tasks are very easy being converted to verifiable rewards.

**Questions:**

Please see the weakness.

---

> ### Author Response · Authors · 2025-11-24
>
> We thank Reviewer 1F8A for recognizing our method's simplicity, practicability, and cross-domain generality, as well as the value of our qualitative analysis. The insightful questions help us improve the manuscript. Below, we summarize and address your concerns.
>
> ----
>
> **Q1.** The baseline coverage is limited: there are no direct comparisons with strong RLHF/DPO or verifier-driven multimodal RL training pipelines, making it hard to judge competitiveness. Also, the improvements on some cases are very marginal.
>
> **A1.** To contextualize the performance of our SSL4RL method, we include a strong, task-specific baseline VLM-R1 [1]. It extends R1-style reinforcement learning to VLMs with rule-based reward formulation. Specifically, we randomly split downstream datasets into 60% for training and 40% for held-out testing. Then we fine-tune Qwen2.5-VL-3B using the GRPO algorithm, where the reward is computed directly based on golden answer. We denote the tuned model as ***Golden-3B***. Notably, the model is trained with **oracle reward signals**, and the setup simulates an ideal scenario where the reward model is perfectly aligned with the evaluation metric.
> The results are presented in ***Table R1*** and ***Table R2***, where we draw two key conclusions:
> - **The Oracle Baseline has a high but finite ceiling.** Even with direct supervision from golden answers, the Golden-3B model achieves 84.93% on MMBench and 73.21% on SEED-Bench. This indicates inherent challenges in these benchmarks that are not fully solved even with ideal, verifiable rewards.
> - **Our SSL4RL method is highly competitive.** The performance gap between our best SSL4RL variants and the Golden-3B oracle is relatively small (e.g., 81.35% vs. 84.93% on MMBench, and 69.80% vs. 73.21% on SEED-Bench), compared to our improvements over the base model. This demonstrates that our self-supervised objectives, which require **no labeled downstream data**, can effectively close the gap to the performance driven by idealized, task-specific reward signals.
>
> This comparison paints an encouraging picture for SSL4RL. Our method, which deliberately avoids using any downstream labels, can close much of the gap to a model trained with direct oracle signals. While perfect verifiable rewards remain a powerful tool, our work shows that highly competitive performance can be achieved through a more scalable and generalizable pathway.
>
> **Table R1. Performance (%) of SSL4RL models on MMBench test set.**
> | Category | Model | Logical | Relation | Attribute | Coarse | Cross-Inst. | Single-Inst. | Average |
> | -| - |- | - | - | - | -| -| -|
> |Base model | Qwen2.5-VL-3B| 53.03 | 46.19 | 77.10 | 74.56 | 67.94 | 81.41 | 72.63 |
> | RL with golden rewards | Golden-3B | 71.82 | 86.70 | 88.34 | 83.74 | 79.57 | 89.39 | 84.93 |
> | SSL4RL-3B | Rotation | *66.06* | **82.97** | **87.38** | *80.07* | **75.07** | *85.93* | **81.35** |
> | ~ | Jigsaw | 62.53 | *79.06* | *85.84* | 77.39 | 66.43 | 84.14 | 78.46 |
> | ~ | Contrastive | 60.50 | 71.50 | 73.75 | 66.31 | 56.17 | 77.72 | 68.82 |
> | ~ | Position | **67.58** | 78.37 | 85.16 | **81.22** | *70.70* | **87.87** | *81.12* |
>
> **Table R2. Performance (%) of SSL4RL models on SEED-Bench test set.**
> | Category | Model |  TU | VR | SU | IId | IIn | IA | IL | SR | IC | Average |
> | - | - | - | - | - | - | - | - | - | - | - | - |
> | Base model | Qwen2.5-VL-3B | 59.04 | 60.44 | 67.17 | 63.83 | 61.80 | 51.28 | 57.20 | 59.09 | 59.54 | 62.00 |
> |RL with golden rewards | Golden-3B | 67.68 | 68.41 | 74.76 | 77.03 | 74.69 | 82.05 | 63.84 | 77.27 | 77.10 | 73.21|
> | SSL4RL-3B | Rotation | *62.60* | *62.64* | *72.28* | 73.69 | *71.28* | *48.46* | 55.35 | **68.18** | **77.10** | *69.01* |
> | ~ | Jigsaw | 60.16 | 60.99 | 71.31 | *74.01* | 69.53 | *48.46* | **59.78** | 63.64 | *74.81* | 67.97 |
> | ~ | Contrastive | 55.08 | **64.01** | 69.79 | 68.76 | 61.59 | 30.77 | 56.09 | 56.82 | 74.05 | 63.02 |
> | ~ | Position | **63.21** | 62.36 | **72.83** | **74.17** | **72.40** | **58.97** | *57.93* | *65.91* | 73.28 | **69.80** |
>
> [1]. Shen, Haozhan, et al. "Vlm-r1: A stable and generalizable r1-style large vision-language model."  arXiv:2504.07615 (2025).
>
> **NOTE for Revised PDF: The experiments are added to Section 4.1 and discussed in Appendix G in detail.**

---

> ### Author Response · Authors · 2025-11-24
>
> **Q2.** The 7B results show small gains, supporting a capacity–difficulty mismatch; the method’s effectiveness at modern larger scales remains unclear without harder SSL tasks.
>
> **A2.** Beyond the four SSL4RL tasks, we further explore two more challenging tasks on 7B models:
> - **Mask.** Guided by the insights from Masked AutoEncoders (MAEs), we adopt the self-supervised strategy of mask-and-reconstruct. We randomly sample four patches from an image and mask one of them in the original view. The model must then identify which of the four patches is missing.
> - **Hard-Negative Contrastive.** Moving beyond the simple binary task Contrastive, we increase the discrimination difficulty by introducing hard negatives. For a given image, we use a pretrained DINOv2  encoder to find its most semantically similar images from the dataset. The model's task is to identify the positive augmented view from two hard negative views, forcing it to learn finer-grained visual distinctions.
>
> As shown in **Table R3**, on MMBench, our SSL4RL-7B models surpass the base model by an average of 1.41 points (87.78% vs. 86.37%), and **the Hard-Negative task drives a 5.23% improvement on the Logical Reasoning sub-task.**
>
> Crucially, **extending our evaluation to four additional benchmarks reveals even larger improvements for the SSL4RL-7B models**.  As shown in **Table R4**, our SSL4RL models outperform the base model by **a significant margin of 5.05% on average (62.61% vs. 57.56%)**, particularly on complex VQA benchmarks like BLINK (+6.73%) and V* (+5.76%). These results are consistent with the positive findings on 7B models in concurrent works [2, 3], verifying that our SSL4RL strategy is effective and has potential when scaled to larger models.
>
> **Table R3. Test performance (%) of SSR4RL 7B models on MMBench downstream tasks.**
> | Category | Model | Logical | Relation | Attribute | Coarse | Cross-Inst. | Single-Inst. | Average |
> | - | - | - | - | - | - | - | - | - |
> | Base | Qwen2.5-VL-7B | 76.49 | *84.68* | 85.69 | *84.66* | 84.49 | 89.15 | 86.37 |
> | SSL4RL-7B | Mask | *80.16* | 83.52 | **86.94** | 84.50 | *84.89* | *89.81* | *86.85* |
> | ~ | Hard-Contrastive | **81.72** | **85.71** | *86.21* | **85.43** | **86.17** | **90.98** | **87.78** |
>
> **Table R4. Test performance (%) of SSL4RL 7B models on extended VQA benchmarks.**
> | Category | Model | V* | RealWorldQA | BLINK | MME-RealWorld-Lite | Average |
> | - | - | - | - | - | - | - |
> | Base Model | Qwen2.5-VL-7B | 73.29 | 65.88 | 45.50 | 45.59 | 57.56 |
> | SSL4RL-7B | Mask | *78.01* | *68.88* | *49.28* | **49.03** | *61.30* |
> | ~ | Hard-Contrastive | **79.05** | **70.58** | **52.23** | *48.61* | **62.61** |
> | Maximal Improvement | ~ |  ↑ 5.76 | ↑ 4.70 | ↑ 6.73 | ↑ 3.44 | ↑ 5.05 |
>
> [2]. Wang, Zifu, et al. "Jigsaw-R1: A Study of Rule-based Visual Reinforcement Learning with Jigsaw Puzzles." arXiv:2505.23590 (2025).
>
> [3]. Liu, Yuhong, et al. "Spatial-SSRL: Enhancing Spatial Understanding via Self-Supervised Reinforcement Learning." arXiv:2510.27606 (2025).
>
> **NOTE for Revised PDF: The above discussions are added to Appendix J.**
>
> ----

---

> ### Author Response · Authors · 2025-11-24
>
> **Q3.** Since the RL training is often unstable, it would be good to see the training dynamics such as loss curve and entropy.
>
> **A3.** Thanks for the constructive question. We provide the rewards, entropy, and response length trajectories during RL training in **Appendix K of the revised paper**.
>
> **For rewards**, the four tasks present **stable climbing trends and converge within 300 steps**. Position and Rotation tasks exhibit gradual learning curves, plateauing at a reward of about 0.8 after approximately 200 epochs. For the harder Jigsaw task, the reward surges and plateaus within 20 epochs for 3B models, while 7B models present smoother reward curves, indicating a stronger learning capability of large-scale models. For the Contrastive task, models rapidly surge to 1.0 within 50 epochs due to the task's easy complexity.
>
> **For entropy**, it shows **a decrease-then-increase trend**, an emergent signature of a successful optimization process balancing two competing objectives: ***specialization*** and ***generalization***. At the first stage, the learning rewards incentivize a lower-entropy, specialized policy for high performance on the target task, resulting in a sharpening of the probability distribution. At the second stage, to reduce the costly KL penalty and prevent mode collapse, the policy model slightly broadens probability distributions.. The two-stage entropy curves reflect a sustainable compromise between maximizing reward and preserving the foundational knowledge and generative diversity, supporting our experimental findings that SSL4RL models can generalize well to downstream vision-language tasks.
>
> **NOTE for Revised PDF: The visualizations and discussions are added to Appendix K.**
>
> ---

---

> ### Author Response · Authors · 2025-11-24
>
> **Q4.** Can the authors provide some insights into not-easy-to-verify tasks, since the chosen tasks are very easy being converted to verifiable rewards.
>
> **A4.** SSL4RL requires no human labels, external verifiers, or heuristic judges, yet produces dense and scalable reinforcement signals. Intuitively, it has potential for open-ended tasks where ground truth is ill-defined and human annotations are expensive. To evaluate SSL4RL's potential on open-ended tasks, we leverage **a recent image captioning platform, CapArena [4]**, which contains over 6,000 human-annotated pairwise preference battles. In CapArena, captions from a test model are compared against those from strong baseline models (GPT-4o, CogVLM-19B, or MiniCPM-8B) using a LLM (GPT-4o) as a judge, with human references provided for context. The winner is assigned +1, the loser with -1, and 0 for a draw in each pairwise comparison. **This task mirrors the real-world challenge of improving a model without a single clear and correct answer.**
>
> We evaluate our four SSL4RL strategies on CapArena against the base Qwen2.5-VL-3B model. For easier comparison, we present the final scores after min-max normalization in ***Table R5***. **SSL4RL consistently improves over the base model, with the largest performance gain of 8.14 points (56.45 vs. 48.31).** This clearly shows that the self-supervised rewards provide a meaningful learning signal even in the absence of a verifiable ground truth.
>
> **Qualitative Insights.** We conduct a qualitative comparison in **Appendix Table 27**, shedding light on how SSL4RL enhances open-ended generation:
> - **Enhanced Detail Capture.** SSL4RL models correctly capture more details in the image. For instance, our SSL4RL model correctly identifies a "scoreboard" displaying "1 0 0" and a "disabled persons' sign," which the base model either misinterprets or omits.
> - **Improved Spatial Reasoning.** SSL4RL models tend to use more precise spatial descriptors like "behind the fence", "to the left of", and "the center of the court", aligning well with the spatial reasoning ability required by SSL pretext tasks.
>
> In summary, this experiment demonstrates that SSL4RL is not limited to tasks with easily verifiable rewards. By providing a dense, automated learning signal derived from the data's intrinsic structure, our method successfully improves model performance on the complex, open-ended task of image captioning.
>
> **Table R5. Performance of SSL4RL models on CapArena Platform. GPT-Score: the score compared with GPT-4o. Cog-Score: the score compared with CogVLM-19B. CPM-Score: the score compared with MiniCPM-8B. Average: the average score.**
> | Category | Model | GPT-Score | Cog-Score | CPM-Score | Average |
> | - | - | - | - | - | - |
> | Base | Qwen2.5-VL-3B | 0.00 | 6.48 | 92.96 | 48.31 |
> | SSL4RL | Rotation | **19.15** | 49.30 | 96.48 | 55.28 |
> | ~ | Jigsaw | *8.87* | 54.93 | *98.59* | *55.64* |
> | ~ | Contrastive | 4.96 | *57.95* | **100.00** | **56.45** |
> | ~ | Position | 0.00 | **59.15** | 92.25 | 51.04 |
>
> [4] Cheng, Kanzhi, et al. "Caparena: Benchmarking and analyzing detailed image captioning in the llm era." arXiv:2503.12329 (2025).
>
> **NOTE for Revised PDF: The above experiments are added to Section 4.2 and discussed in Appendix I.2 in detail.**
>
> ---
>
> Thanks for your comments and hope our answers could address your concerns. Please let us know if there is more to clarify. We are happy to address them during the discussion stage.

---

### Public Comment · ~Zifu_Wang1 · 2025-11-17
**Suggestion for Relevant Prior Work**

Thank you for sharing this interesting work! We enjoyed reading it and are glad to see continuous interest in using self-supervised tasks as intrinsic rewards for reinforcement learning.

We are the authors of a recent paper, which we believe is highly relevant to your study but appears to be missing from the discussion.

* Jigsaw-R1: A Study of Rule-based Visual Reinforcement Learning with Jigsaw Puzzles. Wang et al. *Published on arXiv in May/2025, accepted by TMLR in October/2025.*

To the best of our knowledge, our TMLR 2025 paper was the first to study visual pretext tasks, specifically **jigsaw puzzles** and **image rotation**, as a basis for rule-based reinforcement learning to enhance visual reasoning in MLLMs.

Since your SSL4RL framework also centrally investigates these exact tasks (Jigsaw and Rotation) as verifiable RL rewards, we believe a discussion of our paper in the manuscript would strengthen your paper and help contextualize its novel contributions.

This is intended merely as a constructive suggestion. We’re happy to see your novel contributions (for instance, the systematic comparison of multiple SSL tasks and the extension to graph domains ), which significantly enriches this exciting line of research.

Thank you for your consideration!

---

> ### Author Response · Authors · 2025-11-24
>
> Thank you for reaching out and bringing your highly relevant work to our attention! We truly appreciate your kind words about our paper and are glad to hear that it resonated with you.
>
> **We have added discussions about your paper** (*Jigsaw-R1: A Study of Rule-based Visual Reinforcement Learning with Jigsaw Puzzles*) **in the related works of our revised paper.** And we hope for potential opportunities to exchange ideas or collaborate in the future!

---

### Author Response · Authors · 2025-11-24
**Revise Summary**

We thank all reviewers for their insightful and constructive comments. Below, we summarize the main additions to the revised paper.
- In **Section 4.1 Benchmarks**, we evaluate our models on **four additional VQA benchmarks** (V*, Real-WorldQA, BLINK, and MME-RealWorld-Lite). Consistent improvements are observed, especially on V* (+8.90%) and RealWorldQA (+9.55%).
- In **Section 4.1 Results**, we contextualize the SSL4RL method with **a strong baseline VLM-R1**, which is tuned on golden rewards with RL.  The performance gap between SSL4RL variants and the Golden oracle is relatively small (81.35% vs. 84.93% on MMBench, and 69.80% vs 73.21% on SEED-Bench), compared to our improvements over base models. It demonstrates SSL4RL's effectiveness as a method that requires no labeled downstream data.
- In **Section 4.1 Analysis**, we provide **a sub-task specific analysis**, providing insights into how different SSL4RL objectives function for distinct downstream tasks.
- In **Section 4.2**, we evaluate the SSL4RL strategy on **an open-ended image-captioning platform, CapArena**. SSL4RL consistently improves over the base model, with the largest performance gain of 8.14 points. We also provide qualitative insights of the enhanced detail capture and improved spatial reasoning.
- In **Section 4.4**, we provide **a robustness analysis** by perturbing the original benchmarks. SSL4RL shows consistently better robustness across different perturbation levels.
- In **Section 4.5.1**, we conduct **an ablation study on data volume scaling**. We observe a clear performance improvement across almost all subtasks when increasing the training data scale.
- In **Section 4.5.3**, we conduct **an ablation study on base model choice**. We apply the SSL4RL strategy on the base model Gemma3-4B and also observe notable performance improvements, indicating that SSL4RL is a general-purpose principle for VLMs, not an artifact of a particular model family.
- In **Appendix J**, we provide a preliminary exploration of **harder SSL4RL tasks (Mask and Hard-Negative-Contrastive)** for 7B models,  which outperform the base model by a significant margin of 5.05% on average, particularly on complex VQA benchmarks like BLINK (+6.73%) and V* (+5.76%).

---

### Note · Program_Chairs · 2026-01-17
**Submission Desk Rejected by Program Chairs**

The following references in this submission do not refer to real documents and/or have major errors in bibliographic information:

 Xiaodong Liu et al. Rpt: Reinforced pre-training of large language models. arXiv:2505.07185, 2025a. URL https://arxiv.org/abs/2505.07185.